# Magnetite-Arginine Nanoparticles as a Multifunctional Biomedical Tool

**DOI:** 10.3390/nano10102014

**Published:** 2020-10-13

**Authors:** Victoria E. Reichel, Jasmin Matuszak, Klaas Bente, Tobias Heil, Alexander Kraupner, Silvio Dutz, Iwona Cicha, Damien Faivre

**Affiliations:** 1Max Planck Institute of Colloids and Interfaces, Department of Biomaterials, Am Mühlenberg 1, 14476 Potsdam, Germany; victoria.reichel@univ-paris-diderot.fr (V.E.R.); klaas.bente@bam.de (K.B.); tobias.heil@mpikg.mpg.de (T.H.); 2Laboratoire “Matière et Systèmes Complexes” (MSC), UMR 7057 CNRS, Université Paris 7 Diderot, 75205 Paris CEDEX 13, France; 3Section of Experimental Oncoclogy and Nanomedicine (SEON), ENT Department, University Hospital Erlangen, Friedrich-Alexander-Universität Erlangen-Nürnberg, Glückstraße 10a, 91054 Erlangen, Germany; jasmin.matuszak@charite.de (J.M.); iwona.cicha@uk-erlangen.de (I.C.); 4Department of Anesthesiology, Kurume University Hospital, Cognitive and MolecularResearch Institute of Brain Diseases, Kurume University, 65-1, Asahimachi, Kurume 830-0011, Japan; 5Bundesanstalt für Materialforschung und -prüfung, Unter den Eichen 87, 12205 Berlin, Germany; 6nanoPET Pharma GmbH, Luisencarrée Robert-Koch-Platz 4, 10115 Berlin, Germany; alexander.kraupner@nanopet.de; 7Institute of Biomedical Engineering and Informatics, Technische Universität Ilmenau, PF 100565, 98684 Ilmenau, Germany; Silvio.Dutz@tu-ilmenau.de; 8Aix-Marseille Université, CEA, CNRS, BIAM, 13108 Saint Paul lez Durance, France

**Keywords:** iron oxide, nanoparticle, theranostics, MRI, hyperthermia

## Abstract

Iron oxide nanoparticles are a promising platform for biomedical applications, both in terms of diagnostics and therapeutics. In addition, arginine-rich polypeptides are known to penetrate across cell membranes. Here, we thus introduce a system based on magnetite nanoparticles and the polypeptide poly-l-arginine (polyR-Fe_3_O_4_). We show that the hybrid nanoparticles exhibit a low cytotoxicity that is comparable to Resovist^®^, a commercially available drug. PolyR-Fe_3_O_4_ particles perform very well in diagnostic applications, such as magnetic particle imaging (1.7 and 1.35 higher signal respectively for the 3rd and 11th harmonic when compared to Resovist^®^), or as contrast agents for magnetic resonance imaging (R2/R1 ratio of 17 as compared to 11 at 0.94 T for Resovist^®^). Moreover, these novel particles can also be used for therapeutic purposes such as hyperthermia, achieving a specific heating power ratio of 208 W/g as compared to 83 W/g for Feridex^®^, another commercially available product. Therefore, we envision such materials to play a role in the future theranostic applications, where the arginine ability to deliver cargo into the cell can be coupled to the magnetite imaging properties and cancer fighting activity.

## 1. Introduction

Magnetic nanoparticles, in particular iron oxide particles, hold great a promise for environmental, biomedical and clinical applications [1]. They are employed for a variety of in vitro and in vivo applications, including drug delivery, gene delivery, hyperthermia and can be used in magnetic resonance imaging (MRI) as contrast agents. In recent years, various different systems based on magnetic iron oxide nanoparticles and their coatings have been introduced for potential usage in diagnostics and therapeutics [2,3,4,5,6,7,8].

Key characteristics for these applications are the particle dimensions, which directly govern the magnetic properties of the crystal, but also the colloidal stability that affects the aggregation. For example, the heating efficiency in magnetic particle hyperthermia is governed by the mechanism of magnetic energy dissipation for stable single domain nanoparticles (Brownian and Neel relaxation) [9], which correspond to size between about 10 and 100 nm. According to these desired properties, a variety of synthetic magnetic single domain nanoparticles have been designed for magnetic hyperthermia in recent years. To name a few, chitosan oligosaccharide-stabilized ferrimagnetic iron oxide nanocubes [10], iron oxide nanoparticles coated with D-mannitol [11], self-assembled nanostructured cubosomes [12], and magnetic nanoparticles with covalently bound folic acid linked to a copolymer [13]. 

In MRI, the optimal size is slightly different as been theoretically described [14]. Despite slight divergences in dimension, various systems produced for magnetic hyperthermia can also be used as MRI contrast agents [15]. This list includes mono- and multi-core iron oxide nanoparticles [12] and nearly mono-disperse iron oxide magnetic nanocubes [16]. It was shown recently, that multi-core iron oxide nanoparticles are promising candidates for medical applications, such as hyperthermia, MRI and MPI, which is confirmed by a few animal studies up to now [17]. Larger ferrimagnetic nanoparticles with remanent particle magnetization can lead to particle agglomerates, reaching a critical size similar to that of the red blood cell, around 6 µm. This leads to a high risk of occluding blood vessels of a patient and cause dangerous side effects, even when those particles are coated and even in the absence of an external magnetic field. Alternatively, multi-core nanoparticles display a very weak remanence in a zero field and blood clotting is prevented [18].

In general, the above-mentioned nanoparticles are synthesized by high-temperature processes, and/or complex chemical procedures including several steps and non-environmentally friendly conditions. Biomimetic syntheses have recently advanced as a promising approach towards high-value magnetite particles and assemblies formed under green conditions [19]. Examples are the use of proteins from magnetotactic bacteria [19,20,21,22,23], or of synthetic peptides [24,25] as additive of reaction. Along these lines, we have recently designed a synthetic process towards highly monodisperse and colloidally stable magnetite nanoparticles based on a one-pot synthesis at low temperature by the addition of a polypeptide in a modified co-precipitation assay [26]. The obtained particles have dimension similar to that of magnetosomes, particles formed by magnetotactic bacteria [27], the later with high potential for hyperthermia [2,28,29], as well as contrast agent for MRI [14,30], and for MPI [31]. Due to these similarities and supported by recent work on the use of (poly)arginine in medical application in general [32], and as cell penetrating peptide in particular [33], we decided to investigate the biomedical potential of our magnetite-poly-arginine system. Here, we show that indeed, these particles are biocompatible and exhibit effective properties in terms of hyperthermia, MRI and MPI, while being easy to produce.

## 2. Materials and Methods

### 2.1. Nanoparticle Synthesis and Characterization

#### 2.1.1. Synthesis of polyR-Fe_3_O_4_ Nanoparticles

Magnetite nanoparticles were synthesized by co-precipitation in an aqueous solution as previously described [34]. Briefly, ferrous and ferric iron chloride salts form magnetite by hydrolysis with sodium hydroxide within an alkaline milieu. A computer-controlled titration device which consists of a titration unit (Metrohm Titrino 888) equipped with a 5 mL cylinder, a dosing unit (Metrohm Dosimat 805) containing a 1 mL unit and a Biotrode pH electrode was used (Metrohm AG, Filderstadt, Germany). A 50 mL reaction vessel with a thermostat was used and the temperature was kept constant at 25 °C. In addition, the reactor was kept under a controlled nitrogen atmosphere and all solutions were deoxygenated with nitrogen. The reaction vessel was filled with 10 mL of H_2_O MilliQ to which poly-l-arginine (15,000 MW–70,000 MW) was added until the final concentration of 0.1 mg mL^−1^ was reached. Iron (II) chloride tetrahydrate and iron (III) chloride hexahydrate were used in a stoichiometric ratio of magnetite (Fe^II^/Fe^III^ = ½) to prepare a 0.1 M iron solution. A 0.1 M NaOH solution was used for titration. The reaction was started by addition of the iron solution (1 µL min^−1^) to the reaction vessel containing the poly-l-arginine solution under continuous stirring using a mechanical stirrer. The pH value was kept under permanent control at pH 11 using the NaOH titration unit and a pH electrode.

#### 2.1.2. Transmission Electron Microscopy

Scanning transmission electron microscopy (STEM) images were acquired with a double Cs-corrected ARM200F instrument (JEOL, Tokyo, Japan) set to an acceleration voltage of 200 kV. The samples were prepared as follows: 10 µL of the sample suspensions were dropped on a Parafilm slice. A carbon coated copper grid was placed on the drop with the carbon side facing towards the fluid for 15 min. The liquid phase was then removed afterwards using a precision tissue paper and the grids were washed twice by placing them on a drop of H_2_O MilliQ for 5 min.

#### 2.1.3. Dynamic Light Scattering and Zeta Potential

Dynamic light scattering (DLS) was used to characterize the particles in terms of their size and colloidal stability. The mean intensity-weighted hydrodynamic diameter d_h_ was determined using a NICOMP Submicron Particle Sizer Model 370 (Particle Sizing Systems, Santa Barbara, CA, USA). 200 µL of 0.5 mM (Fe) aqueous sample dispersion was filled in a 5 mm glass tube and measured for 5 min.

### 2.2. Biocompatibility

#### 2.2.1. Real-Time Cell Analysis

Real-time cell analysis experiments were performed using a xCELLigence system (RTCA DP Analyzer, Roche Diagnostics, Mannheim, Germany). Human umbilical vein endothelial cells (HUVECs) were isolated from freshly collected umbilical cords provided by the Department of Gynecology at the University Hospital Erlangen. HUVECs at passage 1 were used for all experiments. The experiments were performed in 16-well E-plates (ACEA Bioscience, San Diego, CA, USA), in which the impedance was measured with the help of microelectrodes localized at the bottom of the wells. Background measurements were performed while adding 100 µL of sterile endothelial cell growth medium within the wells. Afterwards, 50 µL of the previously prepared cell suspension at a concentration of 1 × 10^3^ was added by replacing 50 µL of media from each well. The impedance was monitored using the xCELLigence system. The effects of polyR-Fe_3_O_4_ nanoparticles on HUVEC viability were monitored as follows: After the first 24 h of cell growth, 100 µL of additional media, containing polyR-Fe_3_O_4_ particles or Resovist^®^, or without particles were added. Final nanoparticle concentrations of 0, 12.5, 25, 50, 100, 200 and 400 µg ml^−1^ were chosen. The cell growth was monitored every 10 min for 96 h on hexaplicate samples [35].

#### 2.2.2. Live-Cell Microscopy

HUVECs were seeded at a concentration of 1 × 10^3^ cells per well in 96-well plates in 100 µL medium each. At 24 h after seeding, 100 µL of medium with or without Resovist^®^ or polyR-Fe_3_O_4_ particles were added to the wells to the final concentrations of 0, 12.5, 25, 50, 100, 200 and 400 µg mL^−1^. The cell growth was monitored with a live cell-imager (IncuCyte FLR microscope system, Essen Bioscience, Ann Arbor, MI, USA) for 96 h hours. The experiments were performed in hexaplicates [35].

#### 2.2.3. Flow Experiments with Bifurcation Model

Bifurcation flow-through cell culture slides (y-shaped μ-slides, Ibidi^®^, Munich, Germany) were used for direct microscopic studies. Numerical flow simulations identified distinct regions of shear stress: The laminar shear stress region (10.2–10.8 dyne cm^−2^ at a flow rate of 9.6 mL min^−1^) throughout the straight main channel and the region of non-uniform shear stress at the outer walls of bifurcation (shear stress range from ~6.3 dyne cm^−2^ to ~0.5 dyne cm^−2^) [36]. HUVECs were seeded at a concentration of 7 × 10^5^ mL^−1^ within the bifurcating slides and grown until 90–95% confluence, resulting in a cell monolayer. The cells were then perfused using a programmed peristaltic pump (Ismatec, Wertheim, Germany). The slide channel was perfused with medium (with or without polyR-Fe_3_O_4_ particles at 100 μg mL^−1^ and 400 μg mL^−1^) at arterial shear stress for 18 h. The slides were detached from the peristaltic pump system after 18 h and were washed with PBS buffer. Subsequently they were fixed with 4% formalin for 10 min at room temperature and the HUVECs cytoskeleton was stained using an Alexa488-phalloidin (PromoKine, Heidelberg, Germany). For cell nuclei counterstaining, a Hoechst staining (Molecular Probes, Darmstadt, Germany) was used. The experiments were performed in triplicates.

### 2.3. Biomedical Applications

#### 2.3.1. MRI

The potential of polyR-Fe_3_O_4_ particles as contrast enhancement in magnetic resonance was estimated based on their longitudinal and transversal relaxivity values. T1- and T2-relaxation times of aqueous dispersions of polyR-Fe_3_O_4_ at concentrations of 0.5, 0.25 and 0.1 mM Fe were determined at 39 °C using a nuclear magnetic resonance pulse spectrometer (miniSpec contrast agent analyzer mq40, Bruker Biospin, Ettlingen, Germany) running at 0.94 T. The relaxivity R_1_ and R_2_ in [L (mmol s)^−1^] was determined by the slope of the reciprocal of the relaxation time 1/T_i_ in [s^−1^] as a function of the iron concentration c(Fe) in [mmol L^−1^] with the relaxation rate of the pure solvent 1/T_i_(0) [L (mmol s)^−1^] as intercept according to the following equation:$$\frac{1}{T_{i}}=R_{i}*c\left(Fe\right)+\frac{1}{T_{i\left(0\right)}}$$
where *i* = 1, 22.

#### 2.3.2. Magnetic Properties and Hyperthermia

Magnetic properties of liquid and powder samples were determined by means of vibrating sample magnetometry (VSM, micromag 3900, Princeton Measurements, Lake Shore Cryotronics, Westerville, OH, USA). The heating performance of liquid samples was measured within a magnetic field calorimeter as described before [36]. In short, the samples are placed in a solenoid generating an alternating magnetic field of about 410 kHz and field amplitude of 25 kA m^−1^. The resulting temperature increase of the sample due to the magnetization reversal losses was measured using a fiberoptic temperature sensor, as function of heating time. Considering the particles concentration of the sample, the specific heating power of the particles was calculated from the recorded heating curves.

#### 2.3.3. MPI

Magnetic particle imaging (MPI) is a medical imaging modality, where the spatial distribution of magnetic nanoparticles can be determined using a superposition of oscillating and static magnetic fields [10,11]. In contrast to MRI scans, a quantitative signal can be generated and real-time reconstruction speed reached. The fundamental signal generation process relies strongly on the magnetic characteristics of the used tracer, namely its non-linear magnetization curve. The suitability of a tracer for MPI measurements is determined in a magnetic particle spectrometer [12]. To evaluate the MPI efficacy, the magnetic particle spectrum (MPS) of the particle suspension was recorded at a drive field with an amplitude of 25 mT and a frequency of f_0_ = 25 kHz using a commercial magnetic particle spectrometer system (Bruker BioSpin). The MPS of the previously approved contrast agent Resovist^®^ was measured for comparison. Spectra were measured on 0.5 mM Fe suspensions and subsequently normalized to 1 M Fe. The magnetization amplitude ratio of the 3rd and 11th harmonic of sample and Resovist^®^ (s/R) was used for comparison.

## 3. Results 

### 3.1. Physicochemical Characterization of polyR-Magnetite Nanoparticles

The TEM image of the polyR-Fe_3_O_4_ system shown in Figure 1a confirms the previously reported characterization, where the particles were about 40 nm in diameter and exhibited a flower-like morphology seemingly resulting from the aggregation of smaller units [26]. The hydrodynamic diameter, measured by dynamic light scattering (Figure 2), was determined to $\bar{d}$ = 98 nm directly after synthesis. The mean hydrodynamic diameter and the size distribution does not change significantly over time and lies in the range of 100–130 nm after 8 months, which indicates colloidal stability over time as corroborated by the photograph (Figure 1b).

### 3.2. Biocompatibility

The potential toxicity of nanoparticles is of high concern, especially if they are aimed for diagnostic purposes. In case of nanosystems intended for intravascular applications, endothelial cells are the first-contact cells. Therefore, we tested the effects of polyR-Fe_3_O_4_ on the viability of endothelial cells. For this purpose, HUVECs, which are an established model system for the human endothelium, were incubated with polyR-Fe_3_O_4_ particles and with the contrast agent Resovist^®^ as a control [37]. Two complementing methods, real-time cell analysis and live-cell microscopy, where used to assess the biocompatibility of polyR-Fe_3_O_4_ particles with endothelial cells under static conditions. To mimic more precisely the biological situation after intravascular injection of the nanoparticles into the bloodstream, a bifurcating-flow through model was used to further analyze the effects of the tested nanoparticle system on endothelial cell monolayer in a dynamic culture.

#### 3.2.1. Static Cell Viability Assays

The label-free real-time cell analysis method suitable for nanotoxicity studies was used to monitor cell viability. The cell index of untreated, healthy endothelial cells increased continuously over time as seen in Figure 3. When HUVECs were treated with polyR-Fe_3_O_4_, the particles were well tolerated up to concentrations of 50 µg mL^−1^. No significant differences in growth curves were detected in comparison to the control samples up to that concentration. In HUVECs treated with 100 µg mL^−1^ polyR-Fe_3_O_4_, an inhibition of cell growth was detected, with a significantly reduced cell index already after 24 h (Figure 3). 

These values indicate that although the polyR-Fe_3_O_4_ does not induce endothelial cell death at 100 µg mL^−1^, they have a considerable inhibitory effect on cell growth and or attachment strength at this concentration. The treatment with 200 and 400 µg mL^−1^ resulted in a strong and significant decrease of the cell index at 24 h, 48 h and 72 h even below the initial values, which indicates toxicity and cell death. In comparison, the reference agent Resovist^®^ induced a slight growth inhibition at 100 µg mL^−1^ after 72 h and resulted in growth arrest in HUVECs at 200 and 400 µg mL^−1^ (Appendix A).

To validate the results of the real-time cell analysis, live-cell microscopy was performed using the IncuCyte^®^ FLR system. No difference in confluence (i.e., the number of adherent cells relating to the surface area covered by cells), between polyR-Fe_3_O_4_ treated cells and the untreated control was observed at concentrations up to 100 µg mL^−1^ during the treatment for up to 72 h. A decrease in confluence was detected in samples treated with 200 and 400 µg mL^−1^ polyR-Fe_3_O_4_. However, some cell elongation was observed at 50 µg mL^−1^ (Figure 4) and was more pronounced at higher concentrations, which was not observable in Resovist^®^-treated cells (Appendix A). Combining the results of real-time cell analysis and live-cell imaging, it can be concluded that the decrease in cell index induced by 100 µg mL^−1^ of polyR-Fe_3_O_4_ was not due to decreased endothelial cell viability but rather due to decreased cell adherence, as no changes in confluence were observed at this concentration with IncuCyte. The nanoparticle-induced alteration in cell morphology is most likely affecting the cell index, as previously described [29].

#### 3.2.2. Dynamic Cell Viability

In dynamic experiments in vitro, the toxic effects of circulating substances can manifest themselves as endothelial cell shrinking and detachment. The viability and the confluence, as well as the cell morphology and cell-cell contacts upon nanoparticle treatment can thus be analyzed by immunofluorescent staining. Figure 5 shows the effects of circulating polyR-Fe_3_O_4_ particles on endothelial cells grown in the bifurcating flow-through channels. As compared with the dynamic control experiments without any circulating nanoparticles (Figure 5a), treatment of HUVECs with 100 µg mL^−1^ polyR-Fe_3_O_4_ had no effect either on cell numbers nor on morphology under laminar or non-uniform shear stress conditions (Figure 5b). Within HUVECs exposed to 400 µg mL^−1^ circulating polyR-Fe_3_O_4_ for 18 h, a significant decrease in cell numbers under laminar shear stress was observed. This effect was further pronounced under non-uniform shear stress conditions, where the cell detachment areas were observed (Figure 5c). This was additionally confirmed with the confluence quantification for non-uniform and laminar shear stress regions of the channels shown in Appendix A.

### 3.3. Biomedical Applications

#### 3.3.1. MRI Contrast Agents

Colloidal stability of the particles is mandatory for the determination of relaxation times and finally, for their use as MRI contrast agents. Besides the DLS measurements (Figure 2), the concentration-dependent relaxation rates 1/T1 and 1/T2 in Figure 6 showed good linearity within the measured concentration range, which is an indication of non-agglomerating particles and overall colloidal stability. PolyR-Fe_3_O_4_ exhibit relaxivity values of R1 = 8.5 and R2 = 145.7 L (mmol s)^−1^, which yields a R2/R1 ratio of 17.0.

#### 3.3.2. Hyperthermia

To evaluate polyR-Fe_3_O_4_ for their possible application in magnetic hyperthermia, the specific heating power (SHP) of polyR-Fe_3_O_4_ was measured and yielded 208 W g^−1^, recorded at 410 kHz, (altering field frequency) and 25 kA m^−1^ (magnetic field strength). The measured value is comparable to the results of particles development for hyperthermia of other groups and the prepared polyR-Fe_3_O_4_ particles are promising for application in magnetic particle hyperthermia experiments [36].

#### 3.3.3. MPI

Two features of MPS spectra are typically considered as fundamental in order to judge the usability of the measured particles for MPI. The first feature is the total amplitude of the third harmonic and the second is the largest frequency that can be distinguished from the noise. In our measurements, the third harmonic of polyR-Fe_3_O_4_ was 1.7 higher than the signal acquired from Resovist^®^ (Figure 7). Our particles outperformed the control until the seventeenth harmonic, where the signal was about 1.35 higher than that of Resovist^®^ (Figure 7).

## 4. Discussion

### 4.1. Biocompatibility 

Nanoparticles, including polyR-Fe_3_O_4_ exhibit a large surface area to volume ratio and therefore, are considered chemically reactive in comparison to bulk materials [38]. Undesirable reactions could occur upon their administration in vivo, which usually depend on the material composition. Therefore, polyR-Fe_3_O_4_ particles were tested in static and dynamic conditions on cellular system in vitro and indeed showed effects but limited on cells, such as influence in cell adherence and therefore decreasing cell viabilities under static conditions. The viability of primary endothelial cells was not affected up to 100 µg mL^−1^ and cytotoxic effects were only observed when using high nanoparticle concentrations (larger than 200 µg mL^−1^). Consequently, the polyR-Fe_3_O_4_ particles were well tolerated in static conditions at moderate concentrations (up to 100 µg mL^−1^) and they exhibited comparable in vitro biocompatibility as the reference product Resovist^®^. For colon carcinoma cell imaging for example, the amount of Fe needed is only about 4–5 μg/10^6^ cells [39], which is much lower than what is used for the biocompatibility assay we used. In addition, the polyR-Fe_3_O_4_ biocompatibility was even improved under dynamic conditions as compared to static conditions. Whereas the particles at 400 µg mL^−1^ were highly toxic under static conditions, only a minor decrease in cell numbers was observed at this concentration under flow.

A variety of nanoparticles consisting of different materials were previously tested regarding their cytotoxicity and biocompatibility using the same experimental set up [35], [40]. In these static cell culture assays, one type of polymeric nanoparticles showed cytotoxic effects at a concentration of 50 µg mL^−1^ [35]. Among iron oxide nanoparticles, some SPIONs showed a tendency for cell growth inhibition at the same concentration of 50 µg mL^−1^, with negative effects on cell viability or adherence at 100 µg mL^−1^ [35]. In contrast, other types of SPIONs were well tolerated by endothelial cells up to 400 µg mL^−1^ in static conditions [35], indicating that considerable differences in biocompatibility are possible among the same class of nanomaterials, likely depending on their physical characteristics and coating composition. Our polyR-Fe_3_O_4_ nanoparticles showed the first inhibitory effects on endothelial viability from 200 µg mL^−1^ on, which indicates good biocompatibility in the light of clinically-relevant concentrations. In dynamic cell culture conditions, polymeric nanoparticles comprised already negative effects on endothelial cells at a concentration of 100 µg mL^−1^ [35]. In contrast, other particle types (e.g., lipid nanoparticles and iron oxide nanoparticles) were tolerated better under flow as compared to static culture [35]. The polyR-Fe_3_O_4_ nanoparticles showed a comparable promising performance within dynamic cell culture conditions.

### 4.2. Biomedical Applications

Superparamagnetic substances such as iron oxide nanoparticles typically reduce the relaxation times T1 and T2 of hydrogen protons in the surrounding tissue. The performance of polyR-Fe_3_O_4_ nanoparticles as a potential MRI contrast agent was previously compared to Resovist^®^, showing that the R2/R1 ratio of 17 is higher in polyR-Fe_3_O_4_ as compared to that of Resovist^®^ (R2/R1 = 11) [14]. Contrast agents with higher R2/R1 ratios include electrostatically L-DOPA stabilized iron oxide nanoparticles with significantly larger core diameter [14] or magnetosomes with dimensions similar to the nanoparticles studied here. However, these nanoparticles either have to be stabilized in a time-consuming procedure after synthesis, or originate from an expensive and time-consuming growth of magnetotactic bacteria, differing from the rapid approach of our synthetic route.

Magnetite nanoparticles encapsulated in cationic liposomes were clinically approved in Europe for the hyperthermia treatment of glioblastoma [41]. In hyperthermia, the magnetic properties of nanoparticles are transferred into heat and dissipated to the surrounding environment, whereas the released energy of particles is corresponding to a hysteresis loop. It is described that the dissipated heat of a magnetite nanoparticle is dependent upon its size and composition, but furthermore, upon the frequency the temperature of the applied field [42]. The polyR-Fe_3_O_4_ achieve 2.5 times higher specific heating power (SHP) than a previously approved contrast agent ferumoxides (Endorem/Feridex^®^) (208 W g^−1^ vs. 83 W g^−1^) [10]. Several studies suggested, that the heating capacity of magnetic nanoparticles can be improved, by means of variation in size, shape and iron content of nanoparticles. Two types of iron oxide nanoparticles, i.e., spherical and cubic shaped nanoparticles, were investigated in a recent comparative study to evaluate their inductive heating capacities [43]. These two types of nanoparticles were produced in two distinct average volumes and sizes. In case of smaller size of nanoparticles, the cubic nanoparticles were shown to heat better compared to spherical particles. An opposite trend was observed for larger nanoparticles, whereby the cubic nanoparticles showed a better heating capacity that the spherical ones. Furthermore, the more effective heating efficiency of the small sized cubic magnetic nanoparticles could be attributed to the formation of chain-like aggregates and an additionally enhanced anisotropy. The larger sized cubic magnetic nanoparticles showed less efficiency due to the strong aggregation of those magnetic nanoparticles [43]. PolyR-Fe_3_O_4_ showed very promising heating efficiency, probably due to their chain formation as reported for magnetosomes, the biogenic magnetite nanoparticles from magnetotactic bacteria, which show highly increased heating efficiency in comparison to single magnetite nanoparticles [29]. Additionally, polyR-Fe_3_O_4_ nanoparticles comprise an interesting flower-like internal structure within each magnetic nanoparticle. Hence, the particles themselves are more spherical-like than cubic e.g., as it was shown that physical motion contributes to the heating efficiency, spherical shaped particles have advantage over the cubic ones, when dispersed in water [43]. Additionally, it is known that single crystalline nanoparticles exhibit higher heating capacities than polycrystalline nanoparticles [44]. We have previously shown that polyR-Fe_3_O_4_ are single crystalline, even though they possess a very special internal structure [26]. In conclusion, our polyR-Fe_3_O_4_ nanoparticles represent ideal candidates for further investigations in magnetic hyperthermia, as they fulfill all criteria to reach high heating capacities, both theoretically and in practice.

PolyR-Fe_3_O_4_ nanoparticles are not solely promising candidates for MRI contrast agents or magnetic hyperthermia, but they also possess interesting capacities for MPI. In general, the MPI signal originates directly from the tracer, whereby mainly iron oxide nanoparticles are used. The magnetic moment of iron oxide nanoparticles is about eight orders of magnitude larger than that of protons, measured by MRI, whereas with MPI, the detected magnetization is approximately 22 million times stronger than that measured with MRI at 7 T. To be suitable for MPI, nanoparticles must possess particular physical and magnetic properties. The theoretical limit was calculated that must be achieved using nanoparticles with a diameter of around 30 nm [26]. So far, Resovist^®^ was used as the “gold standard” in MPI, but did not succeed in a satisfying signal [26]. The non-ideal MPI signal of Resovist^®^ is related to the large fraction of particles with very small core diameters and low magnetic moments. Our polyR-Fe_3_O_4_ exhibited higher signals in their 3rd (1.7 times) and 11th harmonic (1.35 times) when compared to Resovist^®^. They thus showed all needed features for a promising MPI performance, such as monodispersity, ideal sizes, single crystallinity and the ability of chain formation. Additionally, polyR magnetic nanoparticles are positively charged, which may be of additional advantage as they can be internalized by cells easily [26,45], which would guarantee a better distribution and rapid uptake for various biomedical applications [46].

## 5. Conclusions

In summary, the polyR-Fe_3_O_4_ nanoparticles described here represent promising candidates for biotechnological applications. The particle synthesis is straightforward. We show the particles are biocompatible and perform well as MRI contrast agents and for MPI. In addition, since the polyR-Fe_3_O_4_ are positively charged and show high cell uptake, the particles are particularly suitable for magnetic hyperthermia and direct tumor therapy where the coating can be coupled with specific antibodies, to stimulate transport and uptake as well as recognition of tumor cells. Accordingly, we anticipate that polyR-Fe_3_O_4_ particles will become a model material with high theranostic potential.

## Figures and Tables

**Figure 1 nanomaterials-10-02014-f001:**
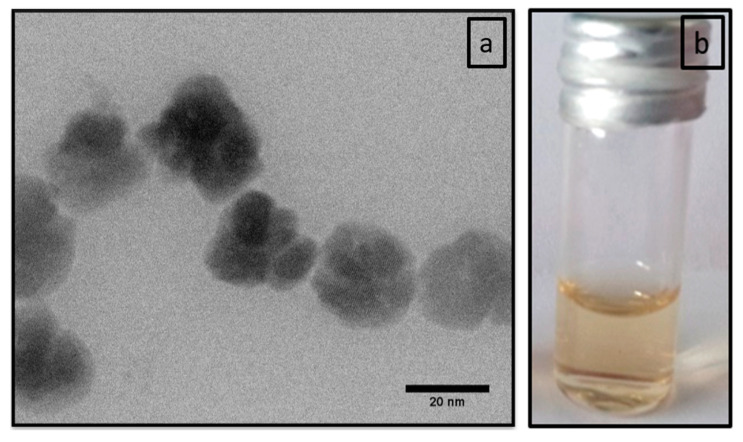
(**a**) STEM image of polyR-Fe_3_O_4_ nanoparticles. (**b**) Image of vial with colloidal stable polyR-Fe_3_O_4_ nanoparticles solution one year after synthesis.

**Figure 2 nanomaterials-10-02014-f002:**
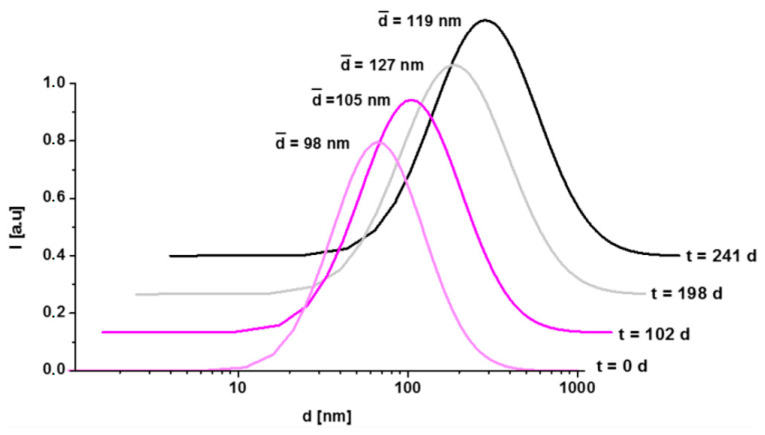
Intensity-weighted log-normal distribution of the hydrodynamic diameter of polyR-Fe_3_O_4_ at different time points after synthesis. The mean hydrodynamic diameter $\bar{d}$ is displayed for comparison.

**Figure 3 nanomaterials-10-02014-f003:**
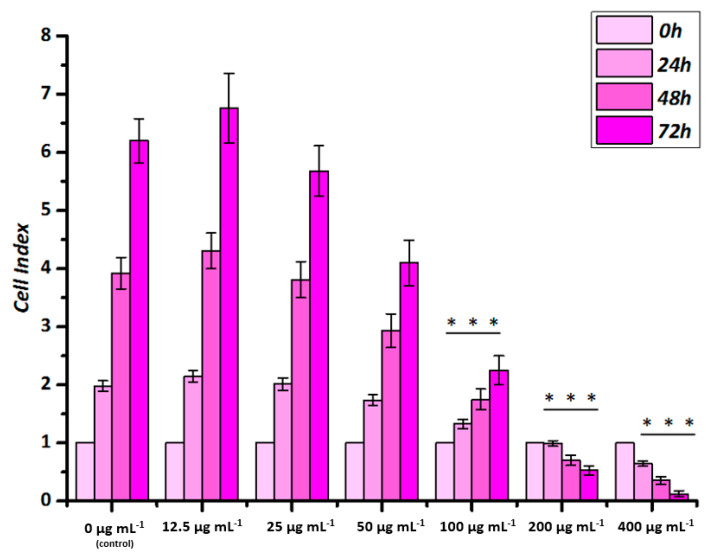
Biological effects of polyR-Fe_3_O_4_ particles on endothelial cells grown in static conditions analyzed by real-time cell analysis. HUVECs were seeded 24 h before nanoparticle application. After the initial 24 h, particles at different concentrations were added and cell index was monitored for up to 72 h post application. Cell Index is displayed as x-fold of untreated controls. Data are expressed as mean ± SEM of n = 3 experiments performed in hexaplicate, *** *p* < 0.001 vs. corresponding control (One-way Anova).

**Figure 4 nanomaterials-10-02014-f004:**
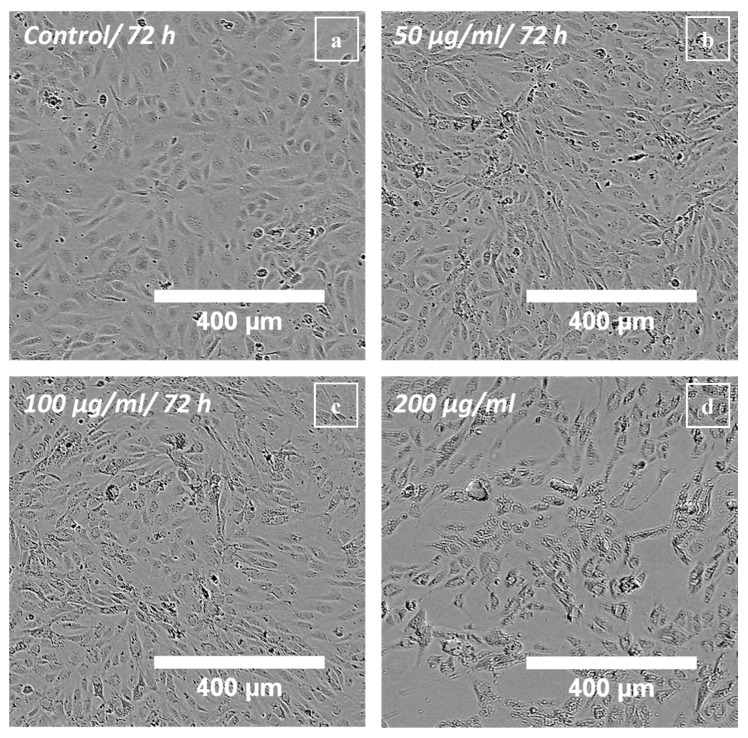
Live-cell microscopy of HUVECs treated with polyR-Fe_3_O_4_: Phase contrast images of HUVECs were taken after 72 h post particle addition. (**a**) Untreated control HUVECs, (**b**) HUVECs treated with 50 µg mL^−1^, (**c**) HUVECs treated with 100 µg mL^−1^ and (**d**) HUVECs treated with 200 µg mL^−1^ polyR-Fe_3_O_4_. Representative images of n = 3 experiments and hexaplicate samples are shown.

**Figure 5 nanomaterials-10-02014-f005:**
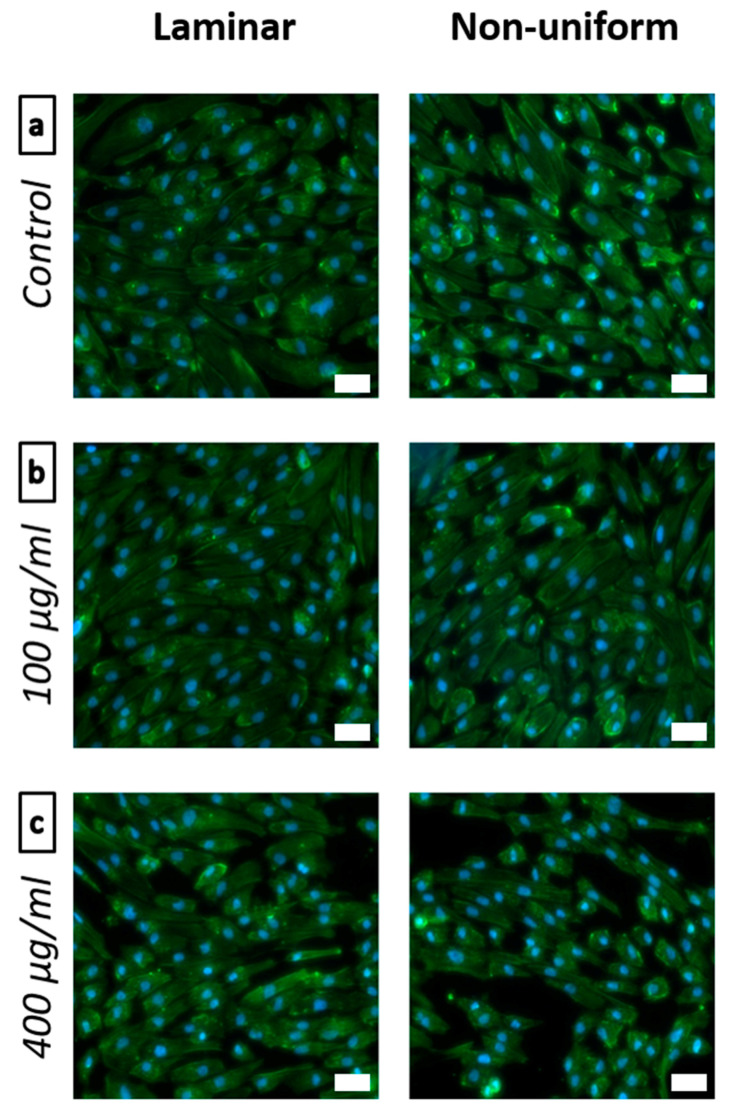
Effects of circulating polyR-Fe_3_O_4_ on enthodelial cell grown under flow conditions. HUVECs were grown in bifurcating slides until confluence and perfused with polyR-Fe_3_O_4_-containing medium for 18 h. The representative laminar and non-uniform regions are shown after fluorescent staining. Nuclei are visualized using a Hoechst 33342 (blue) staining, whereas F-actin is visualized with Alexa 488-conjugated phalloidin (green). Representative images of n = 3 experiments are shown. (Scale bars: 50 μm).

**Figure 6 nanomaterials-10-02014-f006:**
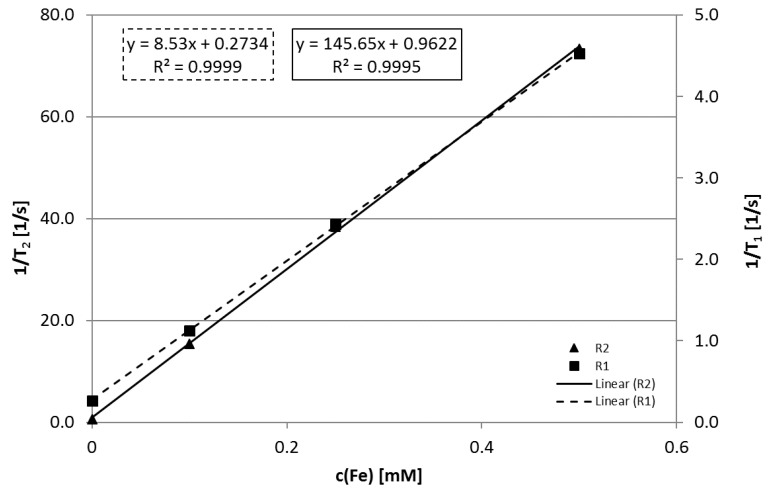
Relationship of the iron concentration c(Fe) in [mM] and the reciprocal of the relaxation time 1/T_1,2_ in [1/s]. The relaxivities R1 and R2 were calculated using the slope of the obtained linear equation.

**Figure 7 nanomaterials-10-02014-f007:**
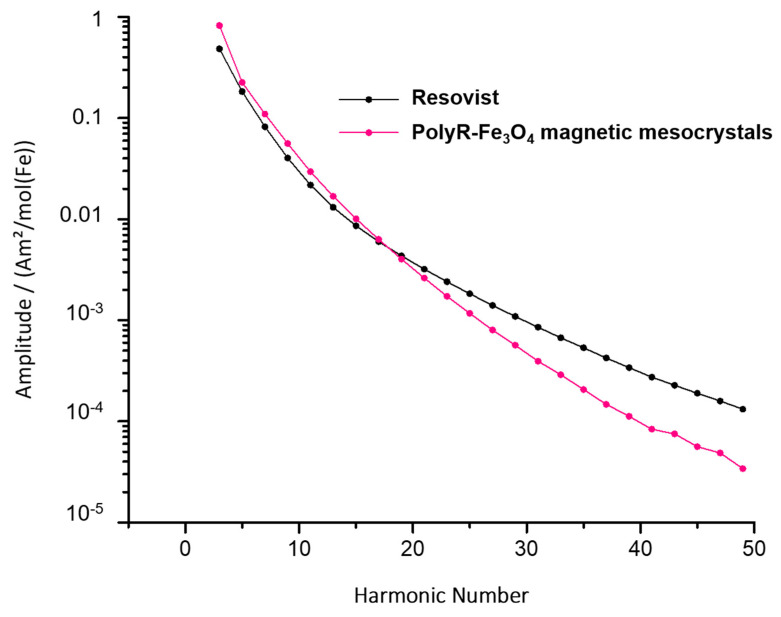
MPS measurement of polyR-Fe_3_O_4_ particles (pink) and Resovist^®^ (black) as comparison.

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
