# Peer review of "Magnetite-Arginine Nanoparticles as a Multifunctional Biomedical Tool"

_nanomaterials, 2020, doi:10.3390/nano10102014_

Round 1
Reviewer 1 Report
In this paper, the authors investigated the biocompatibility and properties of polyR-Fe3O4 particles in terms of hyperthermia, MRI and MPI. The results obtained in this paper show the comparable biocompatibility and better magnetic properties for biomedical applications compared with commercially available particles. The paper is well organized, technically sounds.
I think this paper is accepted after some minor revisions.
Questions and comments:
- Error message (Error! Reference Source Not Found) can be found in both the main manuscript and Supporting Information.
- The scale bar in Figure 5 is almost invisible.
- Caption of Figure 6 : “-1” should be a superior figure.
- Figure 7 : The harmonic amplitudes of the investigated polyR-Fe3O4 particles are larger than those of Resovsit particles for lower harmonics. However, the investigated particles showed the smaller harmonics amplitudes compared to Resovsit particles for higher harmonics. How do the harmonic amplitudes of higher harmonics affect the MPI performance?
Reviewer 2 Report
This paper describes the synthesis, characterization and application of Iron oxide nanoparticles in the biomedical field. Particularly, PolyR-Fe3O4 particles have been utilized for realizing a theranostic approach for cancer treatment based on detection (MRI) and therapy (heating). As the same authors have pointed out, the synthesis of PolyR-Fe3O4 particles has been previously reported and the main claim (in terms of novelty) is their utilization in the biomedical field. Said that, several concerns (mandatory major revisions) must be addressed before the paper can be considered for publication:
1) Figure 1a shows the STEM image of PolyR-Fe3O4 particles. It seems that the average diameter is about 20 nm. However, Figure 2 shows that the average hydrodynamic diameter starts from 98nm. Please clarify this very important aspect;
2) Figure 3 clearly shows that the biocompatibility of PolyR-Fe3O4 particles is high only for very low concentration (12.5 ug/mL). It means that PolyR-Fe3O4 particles exhibit a very high toxicity for higher concentrations. I suggest addressing this point throughout the text and remove the statement that PolyR-Fe3O4 particles are biocompatible;
3) Include the absorption spectrum of PolyR-Fe3O4 particles;
4) I suggest reporting the temperature variation under the influence of the magnetic field. Also, provide the temperature variation as a function of the PolyR-Fe3O4 particles concentration. It is important to show the temperature increase at 12.5 ug/mL;
5) Section 4.2 – biomedical applications is completely meaningless. This section is not related to the scientific value of the manuscript.
Reviewer 3 Report
This article reports poly-L-arginine-coated magnetite nanoparticles and authors claim that the nanoparticles exhibit similarly low cytotoxicity as Resovist, carboxydextran-coated magnetite, but showed higher imaging capability. The concept is not novel and there are many papers reporting polyarginine or peptide-coated magnetite already published. The data shows that the arginine magnetite 1.7 and 1.35 higher in the MR signal compared to Resovist, but there is no evidence of how much this is significant in improving the quality of the MR images. Moreover, the polyarginine magnetite seems more toxic than Resovist at the concentration higher than 200 ug/mL. Authors envision this particle can also deliver cargo into the cells, but not sure about how much it would be better than Resovist in terms of endocytosis, drug loading, and release unless they provide any data.
Round 2
Reviewer 3 Report
Authors did not address the raised issues and could not meet reviewers' satisfaction and/or journal quality.
Author Response
We performed English Proofreading